# Context-specific regulation of surface and soluble IL7R expression by an autoimmune risk allele

Hussein Al-Mossawi [1], Nicole Yager[1], Chelsea A. Taylor[2], Evelyn Lau[3], Sara Danielli[2], Jelle de Wit [1], James Gilchrist[3], Isar Nassiri[2], Elise A. Mahe[2], Wanseon Lee[3], Laila Rizvi[1], Seiko Makino[3], Jane Cheeseman[4], Matt Neville [4], Julian C. Knight [3,5], Paul Bowness[1,5] & Benjamin P. Fairfax [2,5]*

IL-7 is a key factor in T cell immunity and common variants at *IL7R*, encoding its receptor, are associated with autoimmune disease susceptibility. *IL7R* mRNA is induced in stimulated monocytes, yet a function for IL7R in monocyte biology remains unexplored. Here we characterize genetic regulation of IL7R at the protein level in healthy individuals, and find that monocyte surface and soluble IL7R (sIL7R) are markedly induced by lipopolysaccharide. In monocytes, both surface IL7R and sIL7R expression strongly associate with allelic carriage of rs6897932, a disease-associated *IL7R* polymorphism. Monocytes produce more sIL7R than CD4 + T cells, and the amount is additionally correlated with the expression of *DDX39A*, encoding a splicing factor. Synovial fluid-derived monocytes from patients with spondyloarthritis are enriched for IL7R+ cells with a unique transcriptional profile that overlaps with IL-7-induced gene sets. Our data thus suggest a previously unappreciated function for monocytes in IL-7 biology and IL7R-associated diseases.

---

[1] Nuffield Department of Orthopaedics Rheumatology and Musculoskeletal Sciences, University of Oxford, Oxford, UK. [2] Department of Oncology, Weatherall Institute of Molecular Medicine, Oxford, UK. [3] Wellcome Centre for Human Genetics, University of Oxford, Oxford, UK. [4] Oxford Centre for Diabetes, Endocrinology and Metabolism, University of Oxford, Oxford, UK. [5] These authors contributed equally: Julian C. Knight, Paul Bowness, Benjamin P. Fairfax. *email: benjamin.fairfax@oncology.ox.ac.uk

IL-7 is crucial for lymphogenesis and peripheral T cell homeostasis. Genetic polymorphisms at *IL7R*, the locus encoding the alpha chain of the IL-7 receptor, are associated with predisposition to several autoimmune diseases including Ankylosing Spondylitis, Multiple Sclerosis, and Primary Biliary Cirrhosis[1–3]. A non-synonymous polymorphism within *IL7R* (rs6897932) leads to differential splicing of the 6th exon encoding the transmembrane domain of the receptor, resulting in expression of a transcript that is translated to form a soluble receptor (sIL7R)[1]. The C risk allele is associated with increased circulating levels of sIL7R, and this is thought to prolong the half-life of IL-7[4]. While the cellular source for this receptor has been assumed to be lymphoid, this has not been unambiguously demonstrated, with much of the data derived from cell-lines and mixed cell populations.

An understanding of the cell types and context in which disease risk loci elicit functional activity can provide novel insights into pathogenic mechanisms. Expression quantitative trait locus (eQTL) analysis provides unbiased insights into the genetic determinants of gene expression. In primary monocytes a major eQTL mapping to rs931555 (in linkage disequilibrium (LD) with rs6897932 ($r^2 = 0.75$, $D' = 0.87$) and *cis* to *IL7R*) is uniquely associated with IL7R expression after chronic exposure to LPS[5]. Despite the lymphoid role of IL7R, eQTL have not been observed in T cell datasets[6–8]. Given the large effect size of this eQTL on IL7R gene expression after LPS stimulation, and the high LD with a disease-associated polymorphism, we aimed to further characterize this observation at the protein level in immune cell subsets across a large cohort. We performed parallel experiments in isolated monocytes under the same conditions as our previous eQTL analysis, and in mixed cell populations using flow cytometry.

We find monocytes from healthy individuals robustly express surface protein IL7R and sIL7R after exposure to innate immune activators in a manner dependent upon rs6897932 allele. Exposure of LPS stimulated monocytes to IL-7 is associated with a specific transcriptional profile and this corresponds to the genes differentially expressed in the synovial IL7R+ monocyte subset, characterized by high expression of *TCF7* and *CCL5*. We demonstrate IL7R+ monocytes are comparatively enriched in the synovial fluid of patients with active Spondyloarthritis and, using single-cell RNA sequencing, we find synovial IL7R+ monocytes have a discrete expression profile, with similarly marked upregulation of *LTB*, *TCF7*. and *CCL5*. These data demonstrate that genetic variation at *IL7R* impacts IL-7 biology specifically in the setting of inflammation and suggest a key, hitherto unappreciated, myeloid role in the functional mechanism of disease risk variants.

## Results

**Monocyte activation induces expression of surface IL7R.** Peripheral blood mononuclear cells (PBMCs) and isolated CD14+ monocytes from healthy volunteers were incubated for 24 h with or without LPS and surface IL7R was characterized with flow cytometry (Supplementary Fig. 1). In both preparations treatment with LPS led to pronounced induction of surface IL7R (Fig. 1a, b), demonstrating LPS induction of *IL7R* mRNA is accompanied by readily detectable surface receptor expression. LPS elicits release of multiple cytokines from monocytes in PBMCs, including IL-4 and IL-6, which alter T cell surface IL7R[9]. In keeping with this, we observed CD4+, CD8+ T cells and CD56+ NK cell surface IL7R fell markedly upon stimulation (Fig. 1c). We found high inter-individual variation in IL7R response to LPS, particularly in monocytes, and observed little correlation between cell types (Fig. 1d), indicative of cell-type specific regulation. Notably, while

within-cell type correlation of IL7R+ between treated and untreated lymphoid cells was high, correlation between percentage IL7R+ CD14 in untreated and LPS treated states was weak (CD14+ within PBMC culture: $r = 0.20$, $P = 0.03$; purified monocytes: $r = 0.05$, $P = 0.66$, two-sided $t$-statistic) highlighting the cell-type and context-specificity of IL7R response to LPS. To explore the effect of other innate stimuli we exposed monocytes to Pam3CysK4 and Imiquimod, agonists to TLR1:2 and TLR7 respectively. While Pam3CysK4 induced monocyte IL7R expression in all individuals, the response to Imiquimod was variable and non-significant (Fig. 1e). Since LPS elicits the release of multiple cytokines we reasoned that late IL7R induction may relate to magnitude of early (autocrine) cytokine response. To test this, the expression of monocyte *IL7R* at 24 h post LPS was correlated with the expression of all genes following 2 h LPS in 228 individuals[5]. We noted 382/15421 probes (FDR < 0.05) where the 2 h expression was correlated with 24 h *IL7R* (Supplementary Data 1). Within the top 10 associated genes we found both tumour necrosis factor (*TNF*) and chemokine ligand 5 (*CCL5*) were strongly associated (*TNF*: $r = 0.34$, P adj. $= 2.9^{-4}$; CCL5: $r = 0.36$, P adj. $= 1.1^{-4}$; Fig. 1f, two-sided $t$-statistic). Given the key role for anti-TNF therapies in the treatment of many autoimmune conditions, we tested whether released TNF may drive monocyte IL7R induction in an autocrine manner by incorporating the anti-TNF monoclonal antibody Infliximab in the media with LPS. This significantly reduced IL7R+ monocytes (mean −8.1%; 95% CI: −5.7: −10.5%; $P = 5.6^{-9}$, two-sided paired $t$-test, Fig. 1g, Supplementary Fig. 2). Finally, we directly tested TNF in a further 78 individuals, incubating cells with TNF (10 ng/ml) for 24 h. This showed that, akin to LPS, TNF alone robustly induces the expression of surface IL7R in monocytes (Fig. 1h), while concomitantly reducing T cell IL7R surface levels.

**rs6897932 regulates monocyte cell surface IL7R levels.** In eQTL analysis of stimulated monocytes we observed the most significant association for *IL7R* mRNA expression after exposure to LPS for 24 h with rs931555 ($P = 2.1^{-26}$, two-sided $t$-statistic), which was also detectable after 2 h LPS (Fig. 2a). Interrogating these data further, we found rs6897932 is associated with *IL7R* expression ($P = 8.6^{-14}$) reflecting the degree of LD between these two loci. Conditioning on rs931555 resolves the rs6897932 association ($P = 0.92$, Supplementary Fig. 3), whereas controlling for rs6897932 leaves a residual effect of rs931555 ($P = 7.6^{-11}$), indicating rs931555 marks the functional haplotype at the mRNA level. After completion of flow-cytometry experiments, the cohort was genome-wide genotyped on the Illumina OmniExpress, with imputation using the UK10K dataset[10]. Genetic association with surface expression in purified monocytes post LPS stimulation was analysed. This demonstrated the most significant local association with surface expression of IL7R to rs6897932, with no significant association in the unstimulated samples (Fig. 2b, c). In monocytes gated within stimulated PBMC, a significant association with rs6897932 was similarly apparent (Fig. 2d). Of note, we were unable to observe a genetic association between surface IL7R and rs6897932 in either CD4+ or CD8+ T cells, in the basal (Supplementary Fig. 4) or LPS/TNF-stimulated state (Fig. 2d). We had reasoned that if genetic regulation of IL7R expression in T cells was context dependent then an alternative stimulant to LPS or TNF may be required. To explore this, we tested the effect of T cell-specific stimulation with CD3/CD28 ligation in a subset of samples. This reduced T cell IL7R surface levels while inducing monocyte surface IL7R without evidence of a genetic association (Supplementary Fig. 5). Similar to LPS, rs6897932 genotype was associated with monocyte surface IL7R after TNF, with the disease risk allele showing lower average surface IL7R ($P = 0.0007$,

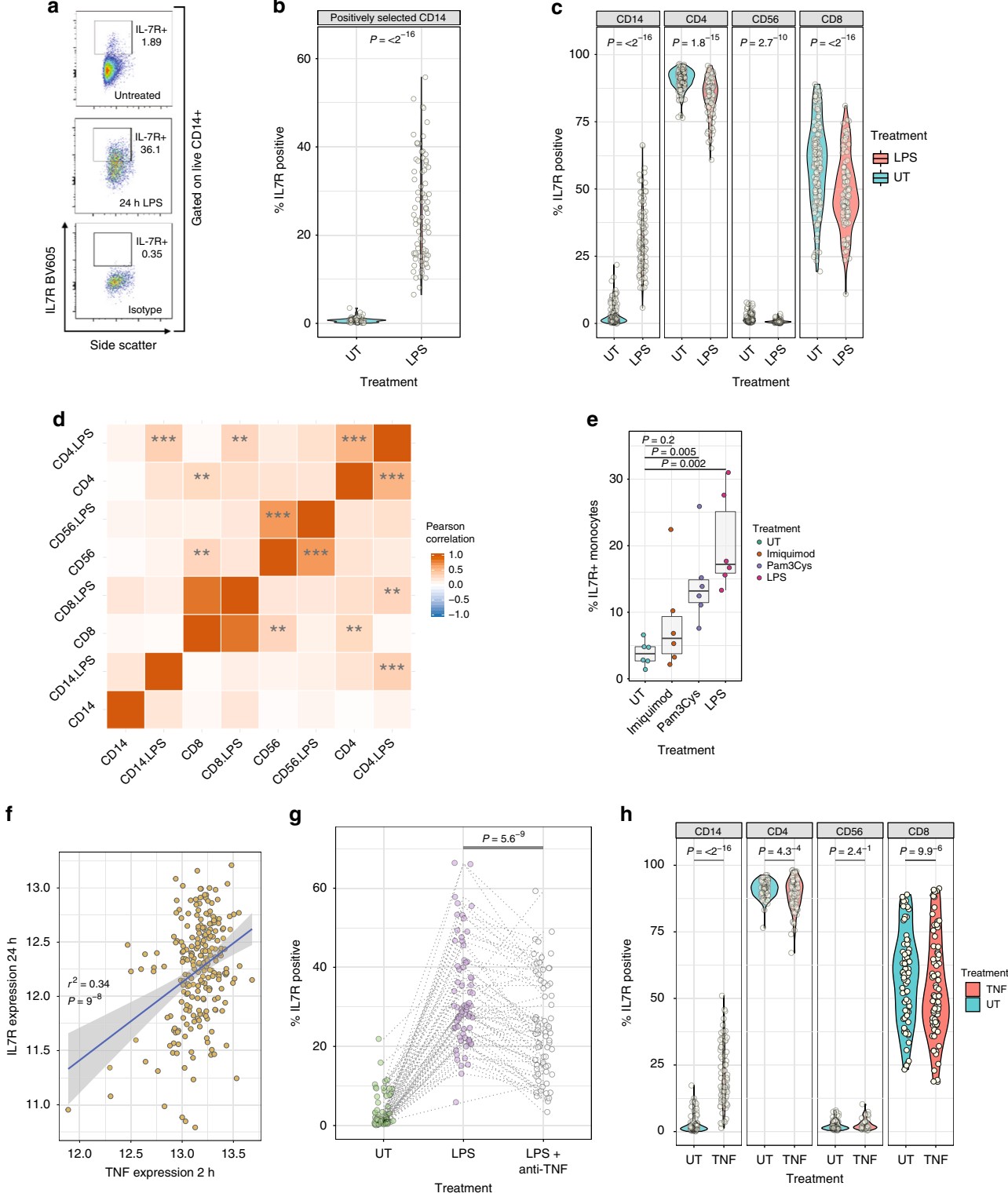

ANOVA, Fig. 2e). These data demonstrate this rs6897932 has a previously unrecognized role in regulating the surface expression of monocyte IL7R after exposure to inflammatory mediators LPS and TNF.

**sIL7R is regulated by rs6897932 and associated with DDX39A.** The reported role of rs6897932 in determining splicing of a soluble form of IL7R in PBMCs[4] led us to investigate whether

stimulated monocytes produce sIL7R and if this is under allelic control. We measured sIL7R on supernatant from randomly selected monocyte samples treated with LPS for 24 h from our previous eQTL study ($n = 161$). Notably, we found monocytes secrete large amounts of sIL7R, the yield being robustly influenced by rs6897932 carriage (mean CC allele = 3149.5 ng/ml, TT allele = 916.9 ng/ml, P allelic effect = $4.7^{-15}$, Fig. 3a). We found only a weak effect of the eQTL locus rs931555 on sIL7R alone ($P = 0.006$, two-sided $t$-statistic), and when controlling for

**Fig. 1** LPS induces profound monocyte IL7R cell surface expression. **a** Representative flow cytometry plots from live gated positively selected CD14 monocytes from one individual where cells were incubated for 24 h alone (top panel) or in presence of LPS (bottom panels). **b** Violin plot demonstrating significant induction of IL7R+ monocytes in paired cultures of positively selected monocytes (Monocyte IL7R+ post LPS- median: 23.8, min: 6.5, max: 55.8, IQR: 15.6–34.9%; $n = 84$, paired $t$-test). **c** Comparative effects of LPS on IL7R+ counts across CD14+ monocytes, CD4+ and CD8+ T cells, and CD56+ NK cells from the same PBMC cultures, untreated or with LPS for 24 h (PBMC IL7R+ post LPS-median: 29.9%, min: 5.9, max: 66.40, IQR: 22.3–40.0; $n = 103$, paired $t$-test). **d** Pearson correlation analyses were performed between indicated cells and treatments on number of IL7R+ cells. ***$P < 0.001$, **$P < 0.01$. **e** Comparative induction of IL7R+ monocytes from 6 individuals with monocytes treated for 24 h with either the TLR7 agonist Imiquimod, TLR1/2 agonist Pam3CysK4 or TLR4 agonist LPS (paired $t$-tests). Error bars show mean, IQR, min and max. **f** Array derived RNA expression of *TNF* at 2 h LPS assayed versus RNA expression of *IL7R* from monocytes from same individuals at 24 h LPS. **g** Comparative incubation of monocytes alone ($n = 69$), with LPS or with LPS + anti-TNF monoclonal antibody (Infliximab). Incubation with TNF antagonist significantly reduces LPS induced IL7R+ monocyte counts (paired $t$-test). **h** Violin plots of responses across cell types of PBMCs treated with TNF alone leads to significant induction of IL7R+ monocytes and significantly reduces CD4 and CD8 T cell IL7R+ positivity ($n = 78$, paired $t$-test). Source data are provided as a Source Data file

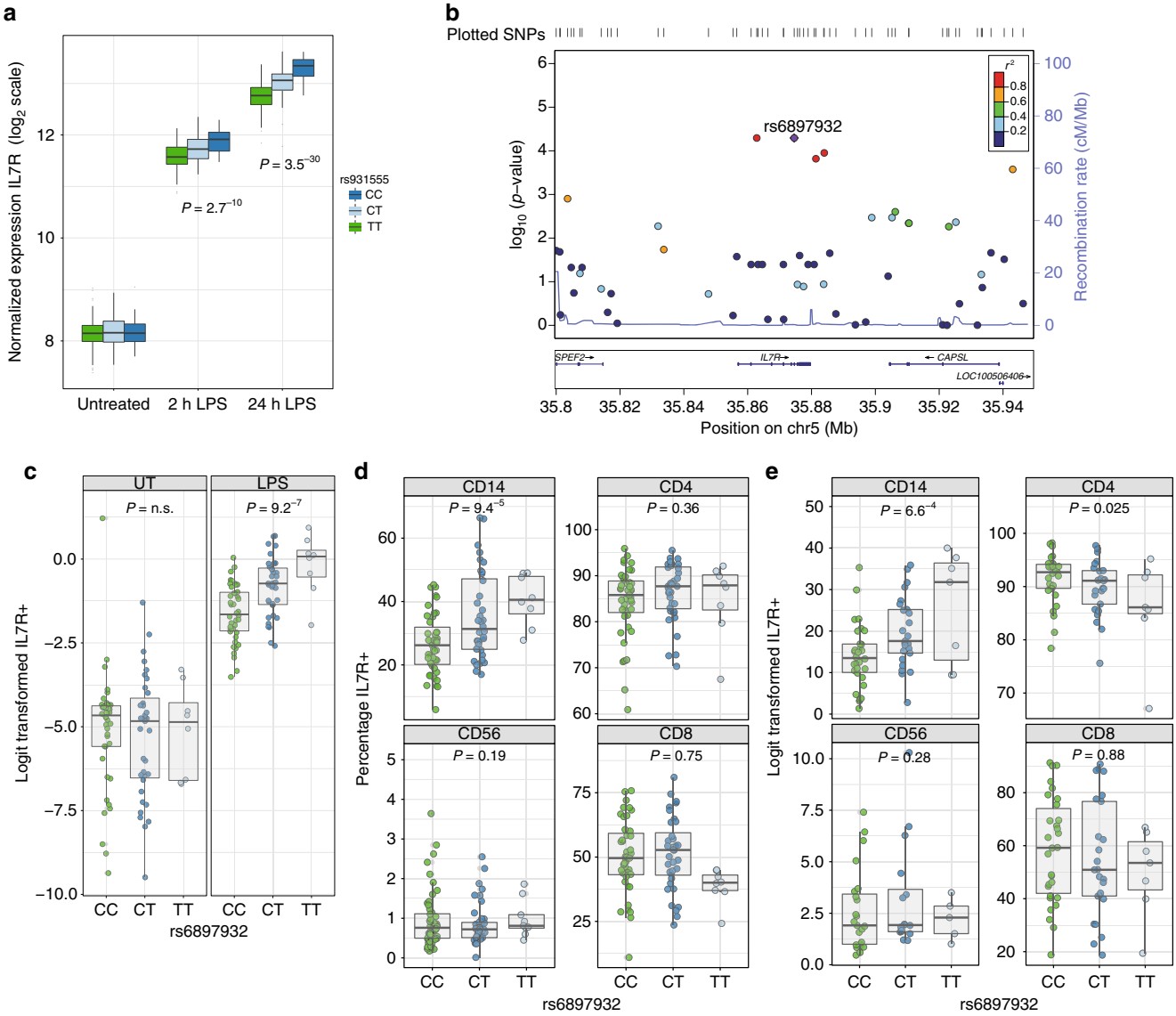

**Fig. 2** rs6897932 specifically regulates monocyte cell surface IL7R levels. **a** Microarray data demonstrating eQTL to *IL7R* noted at rs931555 after both 2 and 24 h treatment with LPS (2 h LPS: $n = 261$, 24 h LPS $n = 322$). **b** Local association plot for monocyte *IL7R* after exposure to LPS from positively selected monocytes demonstrates peak association to rs6897932 ($n = 84$). **c** Batch corrected log values for surface IL7R from positively selected monocytes demonstrating significant effect of rs6897932 carriage after stimulation in comparison to baseline monocytes ($n = 84$, ANOVA for linear model). **d** Surface IL7R in PBMCs across cell subsets after exposure to LPS. ($n = 87$, Note a significant effect of rs6897932 on surface IL7R is only observed in CD14+ monocytes, ANOVA for linear model). **e** As per **d** but PBMCs treated with TNF for 24 h ($n = 62$). Error bars show mean, IQR, min, and max. Source data are provided as a Source Data file

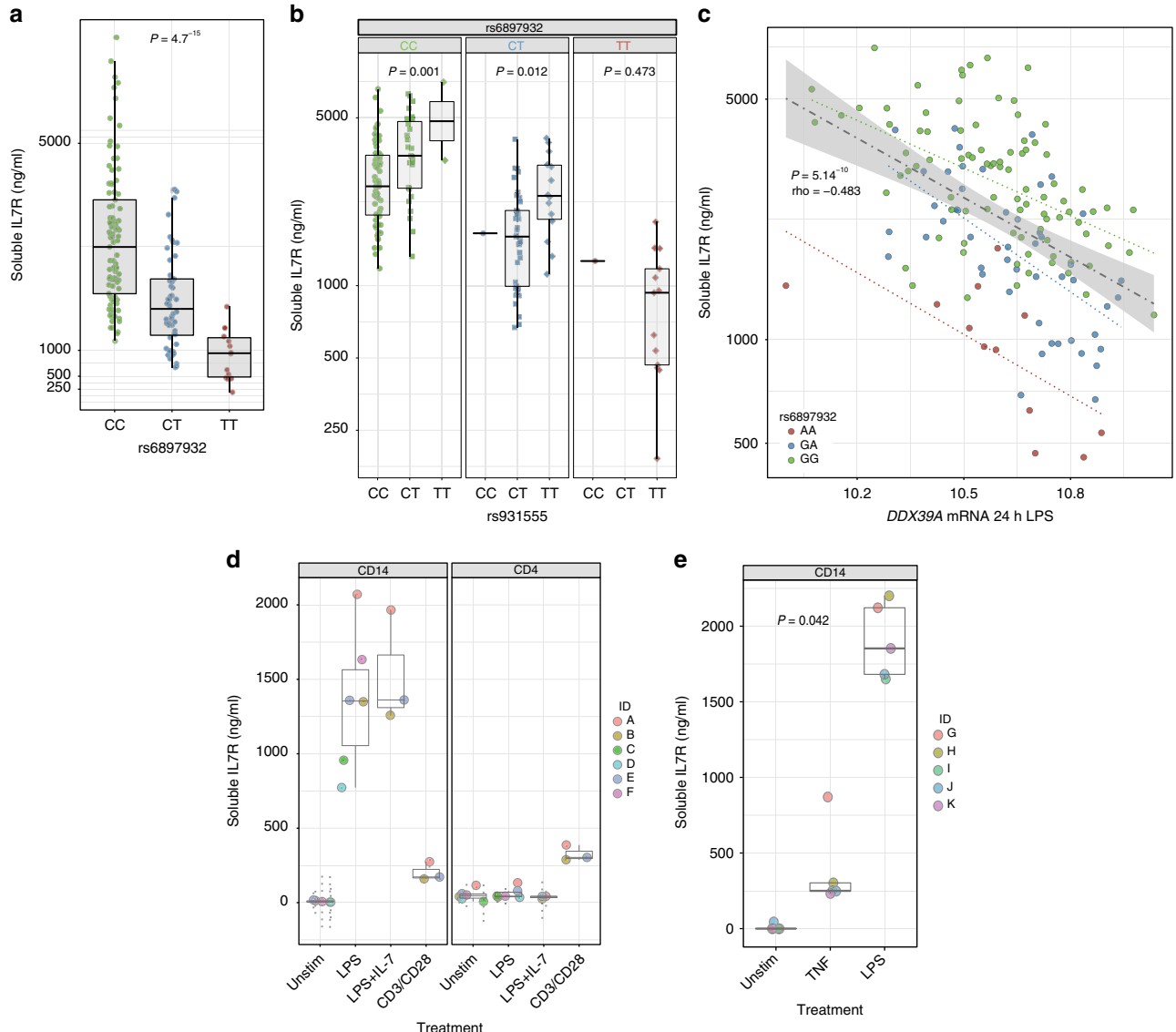

**Fig. 3** Monocyte-derived soluble IL7R is regulated by rs6897932 carriage and *DDX39A* expression. **a** ELISA of sIL7R performed on supernatants of monocytes treated with LPS for 24 h. Samples were randomly chosen from original eQTL dataset and genotypes revealed post-hoc – a significant association was observed at rs6897932 ($n = 161$, ANOVA for linear model). **b** ELISA data from **a** showing additional effect of rs931555 genotype on sIL7R (linear model). **c** Correlation analysis performed between 15421 microarray probes and soluble IL7R (*y*-axis) identified *DDX39A* expression (*x*-axis) to be significantly anti-correlated with soluble IL7R levels, supporting evidence for its regulation of soluble IL7R. Trend lines indicate correlation by allele, with rho and *P*-value for all data ($n = 161$, Spearman rank analysis). **d** ELISA of sIL7R from isolated primary CD14+ and CD4+ cells cultured for 24 h either alone or in the presence of LPS, LPS + IL-7 and CD3/CD28 activating beads. ($n = 6$). **e** ELISA of sIL7R from isolated primary CD14+ cultured for 24 h either alone, in the presence of TNF or in the presence of LPS ($n = 5$, ANOVA). Error bars show mean, IQR, min and max. Source data are provided as a Source Data file

rs6897932 we find rs931555 allele has an additive effect on sIL7R (Fig. 3b), consistent with the upstream eQTL acting independently of rs6897932 to modulate sIL7R.

It has been reported that rs2523506, at the 6p21 gene *DDX39B*, is in epistasis to rs6897932 and that allelic combinations of these two loci further increase the risk of multiple sclerosis[11]. *DDX39B* encodes an RNA helicase that forms a component of the spliceosome[12]. Risk polymorphisms in this gene were associated with increased sIL7R in rs6897932 risk allele carriers[11]. In light of these findings, we proceeded to explore this association in our cohort. We did not observe an effect of rs2523506 on rs6897932 regulation of sIL7R in monocytes (Supplementary Fig. 6), although this may reflect, even in this relatively large cohort, the low frequency of the minor allele at rs2523506 diminishing power to replicate an epistatic effect. To further understand

transcriptional drivers of sIL7R we used pre-existing gene expression data from these samples to explore associations between mRNA expression and sIL7R. In support of the relative independence of soluble protein and total transcript we see no association between total *IL7R* transcript and sIL7R (Supplementary Fig. 7a). We similarly did not observe an association between *DDX39B* expression (at either 2 h or 24 h of LPS – 3 probes to *DDX39B* (BAT1) on the array, all $P > 0.1$) and sILR. However, in strong support of the postulated role of the spliceosome on sIL7R levels, we found a significant correlation between expression of the *DDX39B* paralogue, *DDX39A*, after 24 h LPS and sIL7R, with increased expression of *DDX39A* being associated with reduced sIL7R (2 h LPS *DDX39A* mRNA vs. sIL7R rho = −0.25, $P = 0.01$; 24 h LPS *DDX39A* mRNA vs. sIL7R rho = −0.483, $P = 5.1^{-10}$, two-sided Spearman test, Fig. 3c, Supplementary Data 2). This is

in keeping with the directional effect of *DDX39B* reported in transfected HeLa cells[11]. Total IL7R transcript and *DDX39A* expression were not correlated (Supplementary Fig. 7b), supporting a specific role for *DDX39A* is sIL7R splicing as opposed to expression. Next we tested the relative production of sIL7R in monocytes versus CD4+ T cells, which show the highest levels of IL7R surface expression (Fig. 3d). Whereas neither untreated CD14+ or CD4+ cells release sIL7R, only CD14+ cells release sIL7R in response to LPS, and this was not influenced by IL-7. Conversely, in spite of expressing far higher levels of IL7R basally, CD4+ T cells release very little sIL7R, even when stimulated with CD3/CD28 beads. Finally, we also observed a significant induction of sIL7R from monocytes after TNF stimulation although this was lower compared to LPS (Fig. 3e). These data demonstrate that upon activation, CD14+ monocytes can form a major source of sIL7R.

**IL-7 stimulation activates monocyte transcriptional pathways**. To explore a role for IL-7 signaling in monocytes we co-incubated fresh PBMCs with LPS in the presence of recombinant human IL-7 and performed flow cytometry for a subset of individuals. We repeat previous observations of reduced IL7R expression on T cells[9] upon exposure to IL-7 and find IL-7 similarly reduces monocyte surface IL7R expression, indicating sensitivity of monocyte IL7R to free IL-7 (Fig. 4a). To explore whether ligation of IL7R in monocytes can modulate gene expression, monocytes pre-incubated with LPS to induce IL7R were exposed to IL-7 for 2 h. RNA sequencing was performed on these cells, along with cells treated with LPS alone, from the same individuals. Pairwise differential expression analysis demonstrated this short exposure to IL-7 leads to differential expression of 3240/16186 transcripts tested (FDR < 0.05, Supplementary Data 3 and Fig. 4b, c) including the transcription factor *TCF7*[13], which is required for immature thymocyte survival, the transcriptional activator *MYB*, the non-coding RNA *MALAT1* and known IL-7 regulated cytokines *LTA* and *LTB* which are crucial for the formation of embryological peripheral lymph nodes[14]. Gene ontology analysis of induced genes was consistent with the known anti-apoptotic effects of IL-7, with significant enrichment of genes involved in translation and ribosomal small subunit biogenesis (Supplementary Fig. 8).

To determine whether individual cells might co-express IL7Rmb and sIL7R, while taking into account genotype at rs6897932, we performed single cell quantitative PCR (qPCR) on monocytes ($n = 1818$) isolated from 10 individuals who were homozygous at rs6897932 (4:TT, 6CC). We amplified sIL7R and full-length IL7R, as well as transcripts selected on response to LPS or IL-7, including at baseline and post-exposure to LPS or LPS and IL-7. Applying t-distributed stochastic neighbour embedding (t-SNE) algorithm to these data revealed that a subset of monocytes strongly expressed IL7Rmb post-LPS. Notably, these same cells were responsible for expression of sIL7R and co-expressed *CCL5* and *MALAT1* (Fig. 4d). Using this further cohort of individuals we saw a strong genetic effect of rs6897932 on the distribution of expression of both IL7Rmb and sIL7R expression across cells, with more cells expressing sIL7R and at a higher level in individuals homozygous for the C allele (Fig. 4e). Finally, when we compared cells post-LPS to those post LPS + IL-7 (24 h), we were able to observe a genotype-specific effect of treatment with IL-7 on expression of *MALAT1* (Fig. 4f).

**IL7R+ monocytes form a distinct subset in synovial fluid**. Although our data clearly show the ability of monocytes to induce and express surface IL7R in response to stimulation, the physiological and pathological significance of these observations remains uncertain. To explore the relevance of these cells in an inflammatory disease state, we performed pairwise flow cytometry from blood and synovial fluid in 4 patients with spondyloarthritis (2 Ankylosing spondylitis, 2 psoriatic arthritis). We found that, whereas IL7R+ monocytes were infrequent within PBMCs from these individuals, they were readily observed in the synovial fluid from actively inflamed joints, comprising 12-35% of the monocyte population (Fig. 5a, b). We used single-cell RNA sequencing of paired blood and synovial fluid from a further three individuals with spondyloarthritis to investigate the CD14 myeloid compartment. Unsupervised t-SNE clustering of synovial monocytes revealed 5 distinct clusters (Fig. 5c) with cluster 4 showing high *IL7R* expression (Fig. 5d) relative to the other synovial monocyte clusters identified in the analysis. In addition, the *IL7R* cluster had significantly higher expression of other transcripts including *LTB*, *CCL5* and *IL32* compared to all other synovial monocyte clusters (Fig. 5e). The top 10 genes defining each of the identified synovial monocyte clusters from 5c are shown in Fig. 5f (complete gene list Supplementary Data 4). We secondarily confirmed increased membrane-bound IL7R and sIL7R expression in IL7R+ monocytes using qPCR on sorted synovial CD14+CD11b+ monocytes (Supplementary Fig. 9). Comparison with the *in-vitro* stimulated bulk monocytes demonstrates a significant overlap with the IL-7 responsive genes ($P = 8.5^{-4}$; OR 2.9; 95% CI: 1.5:5.1, two-sided Fisher's exact test). In the paired PBMC monocytes the same clustering parameters only reveal 3 monocyte clusters with no distinct IL7R+ cluster. (Supplementary Fig. 10). Finally, to explore whether the observed IL7R+ myeloid cells might be macrophages, we performed a standard macrophage differentiation assay in primary monocytes from three individuals. We found no induction of IL7R expression in macrophages however, and interestingly these cells became refractory to LPS induced IL7R upregulation (Supplementary Fig. 11). Overall these data confirm that IL7R+ monocytes can be readily observed in humans at sites of inflammation and may represent a unique subset of monocytes based on single-cell gene expression.

## Discussion

The degree to which eQTL correspond to subsequent protein quantitative trait loci is variable[15] and relatively poorly characterised for primary immune cells. The *IL7R* eQTL in monocytes post LPS is particularly intriguing given the large effect size ($r^2 = 0.30$), linkage with a disease associated locus, and the lack of prior exploration of the role of IL7R in monocytes. Here we show that both surface IL7R and sIL7R are robustly expressed at the protein level in stimulated monocytes and the disease-associated polymorphism rs6897932 plays the predominate role in modulating this. As per the eQTL data, the surface receptor QTL is only observed in the stimulated state and is directionally consistent in monocyte and PBMC cultures, reinforcing the reproducibility of these findings. The anti-TNF monoclonal antibody infliximab reduces LPS induction of IL7R. This both implicates autocrine release of TNF, and suggests that our observations may be of pathogenic and clinical significance, since anti-TNF therapy plays a key role in the disease management of the spondyloarthritides. We further confirm that TNF alone can elicit expression of both monocyte surface IL7R and sIL7R. In doing so, we again replicate the allelic effect of rs6897932 on monocyte surface IL7R in a separate cohort upon treatment with TNF. Monocytes act as primary secretors of TNF during acute inflammatory events and so this observation of monocyte-specific feedback resulting in expression of IL7R is of particular interest given the accumulating evidence for myeloid role in many inflammatory disease states[16].

sIL7R is thought to potentiate IL-7 signaling[4] and likely plays an important role in IL-7 biology and disease associations. While

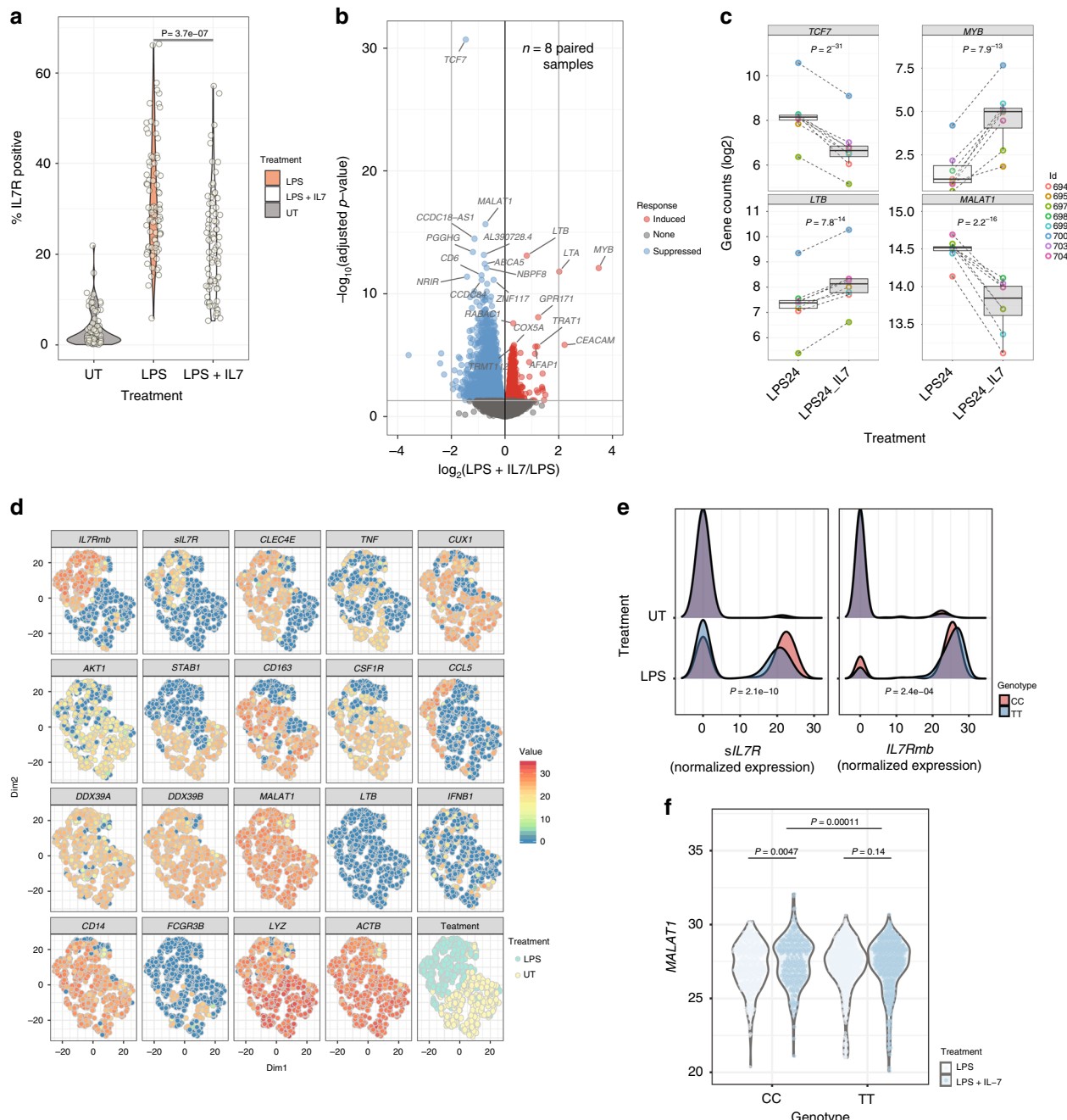

**Fig. 4** Monocyte surface IL7R is functional, activating multiple transcriptional pathways. **a** Violin plot demonstrating monocytes from PBMC cultures in the untreated state, after exposure to LPS and after exposure to LPS with recombinant IL-7. IL-7 leads to marked downregulation of IL7R on LPS stimulated monocytes, indicative sensitivity to exogenous cytokine (paired t-test). **b** Volcano plot of mRNA from RNAseq experiments of 8 paired monocyte samples either treated with LPS alone for 24 h or with LPS with additional IL-7 added for the last 2 h of culture. Treatment leads to widespread differential transcript expression with the most significant 20 transcripts labelled. **c** Example boxplots of genes differentially regulated by recombinant IL-7 in monocytes (linear model). **d** t-SNE plot of single-cell real-time quantitative PCR (RT-qPCR) data from 10 individuals (1218 cells) either untreated or in the presence LPS showing expression of membrane-bound *IL7R* (*IL7R*mb), soluble *IL7R* (s*IL7R*), *CLEC4E, TNF, CUX1, AKT1, STAB1, CD163, CSF1R, CCL5, DDX39A, DDX39B, MALAT1, LTB, IFNB1, CD14, FCGR3B, LYZ,* and *ACTB*. **e** Density plots of single-cell RT-qPCR expression of sIL7R and mbIL7R normalized expression in untreated and LPS treated monocytes according to rs6897932 genotype. Kolmogorov Smirnov test. **f** Violin plots of single-cell RT-qPCR expression of *MALAT1* in LPS stimulated monocytes either alone or in the presence of recombinant human IL-7 according to rs6897932 genotype, t-test

T cells cannot be excluded as a significant source of sIL7R, we find that activated monocytes produce markedly greater sIL7R than either resting or activated T cells and this occurs in a rs6897932-delineated manner, suggesting that in the context of innate immune activity, monocytes can significantly contribute to the local pool of sIL7R. Although we have limited power to

determine epistatic effects, analysis of individuals heterozygous for rs6897932 demonstrates an additional significant allelic effect of rs931555 (in the direction of the previously identified eQTL) on the regulation of sIL7R. It would thus appear that rs6897932 is the primary determinant of sIL7R release from monocytes and is also key in determining surface expression. In keeping with a

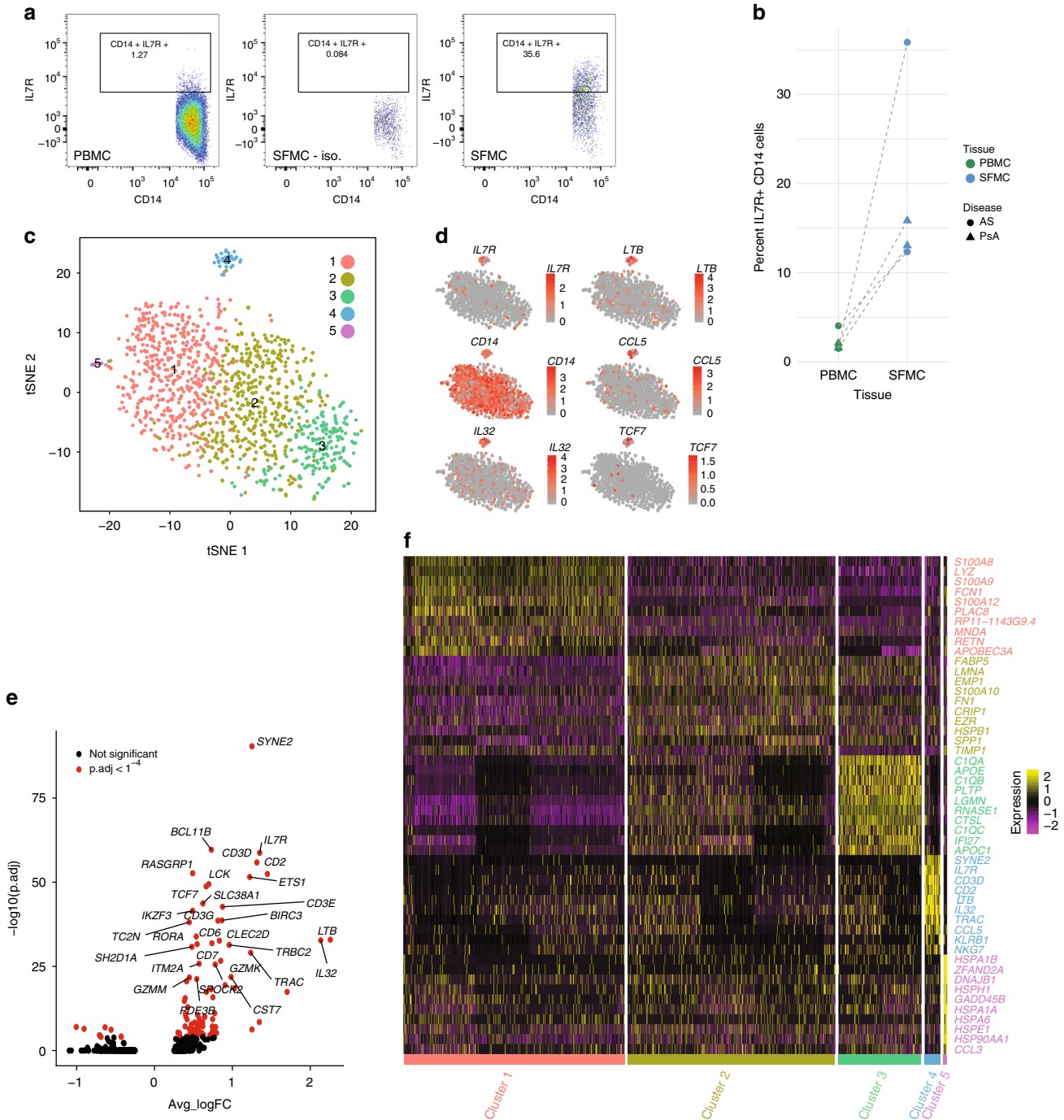

**Fig. 5** IL7R+ monocytes form a distinct subset detectable in synovial fluid. **a** Representative flow cytometry of IL7R staining of spondyloarthritis matched patient PBMC (left) and SFMC (middle isotype control, right) gated on CD3-CD14+ as per sequential gating strategy shown in Supplementary Fig. 1. **b** Results from 4 patients demonstrating comparative monocyte IL7R staining in matched PBMC and SFMC using flow cytometry. **c** t-SNE clustering of monocytes from single-cell RNA sequencing of spondyloarthritis patient SFMC showing 5 clusters (n = 3). **d** Single-cell monocyte expression of IL7R, LTB, CD14, CCL5, IL32, and TCF7 overlaid on t-SNE clustering in **c**. **e** Volcano plot of genes significantly upregulated in the SFMC monocyte cluster 4 compared to all other SFMC monocyte clusters with top 20 genes annotated. Wilcoxon rank sum test. **f** Heatmap of top 10 genes from the each of the five SFMC monocyte clusters identified in **c**

recently described role for the spliceosome component *DDX39B* in IL7R splicing[11], we observed a highly significant correlation between expression of the *DDX39B* paralogue gene, *DDX39A*, and sIL7R levels, with increasing expression of *DDX39A* associated with reduced sIL7R. The absence of association to *DDX39B* may reflect cell type specific mechanisms or potentially a divergence from the steady state induced by stimulation.

We did not observe an effect of rs6897932 on surface expression of IL7R in either CD4+ or CD8+ T cells in resting or stimulated states. Given we did not measure T cell derived sIL7R across the whole genotyped cohort we cannot exclude an allelic effect of rs6897932 on this, but it is notable that stimulated monocyte production of sIL7R is several fold greater than that of stimulated T cells. Interestingly, previous associations of this

allele on sIL7R were from mixed cell populations[4]. Furthermore, there is an absence of an eQTL to IL7R in the GTEX data at rs6897932 and an eQTL is not observed in CD4$^+$ T cells[6]. Whereas sIL7R levels are consistently demonstrated to be raised in autoimmune conditions – notably MS[4], specific attempts to assay sIL7R from primary T cells have not demonstrated an allelic association with rs6897932[17–19]. Our data is thus in keeping with a predominant effect of this allele on sIL7R and also surface protein expression in activated monocytes, and implicates monocytes as a contributory cell type to the disease association.

In addition to the role of monocyte sIL7R in modulating inflammation, the direct ability of monocytes to respond to IL-7 via the surface receptor is intriguing. The single cell sequencing experiments from inflamed joints demonstrates IL7R$^+$ monocytes are readily observable and have a distinct expression profile. Notably, there is a significant overlap between the genes that are modulated in monocytes by exposure to exogenous IL-7, and those differentially expressed within IL7R$^+$ monocyte cluster from the ex-vivo single cell data. While we are unable to dissect splicing effects from the single cell sequencing data, qtPCR data from the same cells demonstrates they similarly express sIL7R. Finally, single-cell qtPCR demonstrates these monocytes are able to co-express both transcripts of IL7R, although this is strongly under control of rs6897932. Of particular interest is the observation that IL-7R$^+$ monocytes express a number of genes previously thought to be T cell specific such as CD2 and LCK, and these data are in keeping with a recently described similar subset from a flow-sorted smart-seq2 data-set[20]. Our data places IL7R$^+$ monocytes at the inflamed joint and suggests these cells may play a hitherto unknown role in inflammation.

IL7R can also dimerize with the thymic stromal lymphopoietin (TSLP) receptor[21,22], however monocyte expression of this is very low and it is not induced by LPS[5], thus we focused this study on investigating the role of IL-7 on monocyte biology. An additional role for sIL7R in the modulation of TSLP signaling cannot be excluded – indeed the UK biobank data shows an association of this allele with eosinophil count[23]. While we demonstrated TNF induces both surface IL7R and sIL7R expression, the degree of this response tended to be less than to that of LPS. This was unlikely to be a dosage effect, as higher concentrations of TNF led to cell death (not shown), but instead suggests a role for other co-released cytokines or the TLR4 pathway directly. Finally, while we did not observe an effect of this allele on other cell subsets, our analysis was of crude CD4$^+$ and CD8$^+$ fractions, and thus allelic modulation of surface IL7R in smaller subsets cannot be discounted.

Notably, our data show a very strong genotypic effect of the disease associated C allele, with higher monocyte derived sIL7R during inflammation. Given the known role of sIL7R in potentiating the half-life and bioavailability of IL-7[4], these results are in keeping with the increased CD8$^+$ T cell counts previously reported in carriers of the C allele[2]. This observation is reinforced by analysis of UK biobank datasets where rs6897932 is associated with lymphocyte percentage[23]. Importantly, this model is in keeping with a role for IL-7 in T cell driven pathogenesis[24] and is compatible with the genetics, where large datasets are unable to demonstrate a direct effect of this allele on T cell expression.

In summary, our study provides unequivocal evidence that the disease associated polymorphism rs6897932 is associated with monocyte specific modulation of surface IL7R and sIL7R in the context of inflammation. We further demonstrate that IL7R-expressing monocytes are responsive to IL-7 with a distinct expression profile that significantly overlaps ex-vivo IL7R$^+$ monocytes from the joints of patients with spondyloarthropathies. These observations highlight a previously unrecognized role for monocytes in IL-7 biology and may have implications for the development of new therapies for autoimmune diseases.

## Methods

**Study participants.** Peripheral blood was obtained from genotyped individuals recruited via the Oxford biobank (www.oxfordbiobank.org.uk) with full ethical approval (Oxford REC: 06/Q1605/55) and written informed consent. Genotype was blinded at the time of study and only revealed at the end of recruitment. Peripheral blood and synovial fluid samples from were recruited with informed consent from patients with inflammatory arthritis attending the Oxford University Hospitals NHS Foundation Trust (Oxford REC: 06/Q1606/139).

**Cell isolation and stimulation.** Whole blood was collected into EDTA tubes (BD vacutainer system) and peripheral blood mononuclear cells were obtained by density centrifugation (Ficoll Paque). CD14 cell isolation was carried out by positive selection (Miltenyi) according to the manufacturer's instructions. Cells were rested overnight (16 h) at 37 °C, 5% CO$_2$ in 5 ml non-adherent polypropylene cell-culture tubes (BD Biosciences) prior to stimulation assays. Cells were stimulated for 24 h in RPMI1640 media with 20% fetal calf serum in the presence of 20 ng/ml LPS (Invivogen), 100 ng/ml pam 3cysk4 (Invivogen), 100 ng/ml imiquimod (Invivogen), 10 ng/ml TNF (R&D systems), 10 ng/ml IL-7 (Peprotech) and CD2/3/28 beads (Miltenyi) at a ratio of 1 bead to 2 cells. In all experiments an unstimulated incubator control was included. TNF blockade was achieved with 5 μg/ml of infliximab (Remicade, Janssen). Macrophages were differentiated in the presence of M-CSF (50 ng/ml) as per previously described[25].

**Flow cytometry.** Staining antibodies and dye clones, dilutions and manufacturer shown in Supplementary Table 1. Cells were stained in phosphate buffered saline containing 1% fetal calf serum on ice and in the dark for 20 min, then fixed in 1.6% paraformaldehyde. All samples included fixable amine reactive viability dye and isotype control for IL-7R. Flow cytometry was performed on a BD Fortessa calibrated daily with calibration and tracking beads from BD Biosciences. Data were analysed using FlowJo software (Treestar®).

**ELISA.** Cell supernatants from stimulated cells were collected and sIL7R quantified as previously described[11]. Briefly, 96-well plates were coated overnight with 1 μg/ml mouse anti-human CD127 mAb (R&D Systems). Plates were blocked with 5% BSA for 1 h, washed, and cell supernatants added for 2 h. Bound sIL7R was detected with 12.5 ng/ml biotinylated goat anti-human CD127 polyclonal Ab (R&D Systems) for 1 h, followed by a 30 min incubation with streptavidin-HRP and a 15 min incubation with TMB peroxidase substrate (Thermo Fisher). The reaction was stopped with H$_2$SO$_4$ and plates read at 450 nm. A standard curve was generated using recombinant human CD127 Fc chimera protein (R&D Systems).

**RNA extraction.** The AllPrep DNA/RNA/miRNA kit (Qiagen) was used for RNA extraction. Cells were spun down and re-suspended in 350 μl of RLTplus buffer and transferred to 2 ml tubes. Samples were then stored at −80 °C for batched RNA extraction. Homogenization of the sample was carried out using the QIAshredder (Qiagen). DNase I was used during the extraction protocol to minimise DNA contamination. RNA was eluted into 35 μl of RNase-free water. The RNA amount was quantified by qubit and the RNA samples stored at −80 °C for storage until ready for sequencing.

**Bulk RNA sequencing.** RNA sequencing was carried out at the Wellcome Trust Centre for Human Genetics core facility. RNA underwent quality control testing using a bioanalyser (RNA 6000 Nano kit, Agilent) followed by cDNA library preparation. Paired end sequencing was performed at 100 base pairs on each side of the DNA fragment on the HiSeq 4000 platform. In total 47–69 million reads were sequenced per sample (mean = 56.5 million) with samples multiplexed to 8 samples per lane.

**Analytical methods for bulk RNA sequencing and graphing.** Bioinformatic analysis of RNA sequencing samples was carried out using validated packages in R. Reads were mapped to human genome reference sequence GRCh38 with HISAT[26]. Gene counts were retrieved using HTseq-count[27] and the Ensembl gene annotation. DESeq2[28] was used for differential expression analysis using a pairwise model where individual was coded for. Pathway and network analysis was performed using the R package XGR[29]. All statistical analysis was performed using base R. Batch correction was performed by incorporating batch within linear regression of logit transformed cell counts. Plots were created using the ggplot2 package[30] and association plots using LocusZoom[31]. For all boxplots the upper whisker extends to the largest value maximally 1.5×IQR from the 75th centile and the lower whisker extends to the smallest value 1.5×IQR from the 25th centile. The junctions of whisker and box (hinge) represents the 25th and 75th centiles while median is indicated by the central line. Outlier values are plotted individually beyond the whiskers.

**Cell preparation for single-cell RNA-seq.** Blood and synovial fluid was collected from patients with peripheral spondyloarthritis and mononuclear cells were isolated immediately by density centrifugation. Cells were then counted and loaded onto a 10× Genomics Chromium machine within 4 h of collection from the patients. Cells were captured and cDNA preparation performed according to the

Single Cell 3′ Protocol recommended by the manufacturer[32] libraries were Sequenced on an Illumina HiSeq 4000 to achieve 75 bp reads.

**Single-cell real-time quantitative PCR.** CD14+ monocytes were collected from healthy donors and cultured as above in the presence or absence of 20 ng LPS (InvivoGen) for 24 h (±10 ng/ml IL-7 (Peprotech)) for 2 h before single cell flouresence activated cell sorting into a 96-well plate. Cells were lysed and reverse transcribed prior to pre-amplification using Superscript III with Platinum taq (Thermo Fisher Scientific) and Superase-IN RNase inhibitor (Ambion). Cells were pre-amplified to amplify any lowly expressed transcripts using Taqman gene expression assays (Thermo Fisher Scientific) and diluted in TE buffer. Single cell qPCR was performed using Taqman Universal PCR Mastermix (Thermo Fisher Scientific), 192:24 integrated fluidic circuits (Fluidigm) and the Biomark HD system following the manufacturer's protocol. Cell-free wells and reverse transcriptase-free controls were included as negative and genomic DNA controls as well as a positive control well containing 100 cells. Data were analysed using Real-Time PCR Analysis software from Fluidigm. Sequences of all primers used are provided in Supplementary Table 2.

**Single-cell RNA-seq analysis.** Sequencing reads were mapped with CellRanger V2.1.0 GRCh38 (ENSEMBL annotation). Quality control, filtering and clustering analysis was carried out with the R package Seurat version 2.3.1. To exclude low quality cells, cells with fewer than 500 genes excluded. Likely doublets were removed by filtering out cells with greater than 2000 genes or 10,000 UMIs. All cells with a mitochondrial fraction greater than 7.5% were also excluded. 8219 PBMC cells and 6668 SFMC cells passed the filtering. Library-size normalization was performed on the UMI-collapsed gene expression values for each cell barcode by scaling by the total number of transcripts and multiplying by 10,000. The data were then natural-log transformed using log1p.

PBMC and SFMCs were clustered separately. 9712 (PBMC) and 9409 (SFMC) genes with high variance we selected using the FindVariableGenes function with log-mean expression values between 0.0125 and 4 and dispersion (variance/mean) between 0.25 and 30. The dimensionality of the data was reduced using principle component analysis and 15 principle components (PCs) were identified for downstream analysis. The Louvain algorithm for modularity-driven clustering, based on a cell–cell distance matrix constructed on the defined PCs was then used. This was implemented using the FindClusters function in Seurat with a resolution of 0.6 to identify 11 distinct clusters of cells in both PBMC and SFMC.

Cluster annotation was assigned using canonical expression markers for key cell types such as CD14 for monocytes. The subset function was then used to extract the raw gene expression matrix of the monocyte cluster for further analysis. The monocyte specific analysis involved re-clustering based 15 principle components and a resolution of 0.6. For PBMC 3 monocyte clusters were identified while in the SFMC compartment 5 SFMC clusters were identified. In the SFMC compartment, the FindAllMarkers function was used to identify the top 20 genes in the SFMC monocyte clusters in order to generate the heatmap. For all single-cell differential expression tests, a non-parametric Wilcoxon rank sum test was used.

**Reporting summary.** Further information on research design is available in the Nature Research Reporting Summary linked to this article.

## Data availability

The raw sequencing data generated for the present study has been deposited in the European Bioinformatics ArrayExpress Archive under the following accession codes: E-MTAB-8225 and E-MTAB-8207. All raw data used for generating figures has been deposited in the Source Data file. Source data for all figures and panels are provided as a Source Data file. All other data are available from the corresponding author upon reasonable requests.

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

## Acknowledgements

We thank the volunteers from the Oxford Biobank (www.oxfordbiobank.org.uk) for their participation and the NIHR Oxford Biomedical Research Centre which supported the recalling process of the volunteers. The views expressed are those of the author(s) and not necessarily those of the NHS, the NIHR or the Department of Health. This work was supported by a Wellcome Intermediate Clinical Fellowship (201488/Z/16/Z to B.P.F.) and The Academy of Medical Sciences Starter Grant for Clinical Lecturers (B.P.F.); Wellcome Investigator Award (204969/Z/16/Z to J.C.K.), Wellcome Grant (090532/Z/09/Z to core facilities Wellcome Centre for Human Genetics; Arthritis Research UK (20773 to J.C.K.) and the National Institute for Health Research (NIHR) Oxford Biomedical Research Centre (BRC) (S.D. and P.B.). H.A.-M. was supported by a Wellcome Studentship (102288/Z/13/Z).

## Author contributions

H.A.-M. and B.P.F. conceived the project. Cell purification and stimulation was performed by E.L., C.A.T., B.P.F., S.M., and S.D. Flow experiments were performed by H.A.-M. with assistance from J.d.W. and S.D. Samples were procured using the Oxford Biobank organized by J.G., J.C., and M.N., and N.Y., E.A.M., and L.R. performed ELISA analysis. B.P.F., H.A.-M., W.L., and I.N. performed statistical and bioinformatic analysis. B.P.F. drafted the manuscript and figures with subsequent contributions from all authors. B.P.F., P.B., and J.C.K. jointly supervised the project.

## Competing interests

The authors declare no competing interests.
