## [Peer Review File · Nature Communications]

Reviewers' comments:

Reviewer #1 (Autoimmunity, innate immunity, cytokine signalling)(Remarks to the Author):

This manuscript describes the presence of IL7R on monocytes after TLR stimulation, which is mediated in part by TNF. The increase appears greater in those with the rs932555 CC allele or the rs6897932 TT alleles, which are in LD. LPS induced DDX39A mRNA expression is related to the sIL7R. Differentially expressed genes are present following IL7 stimulation of LPS treated monocytes showing functionality of the expressed IL7 receptor. Single cell RNAseq is presented for control monocytes and AS/PsA monocytes and their synovial fluid counterparts. Addressing the following issues would strengthen the manuscript.

- 1) which allele of rs6897932 is associated with AS and MS? what is the proposed relation of sIL7 vs monocyte/macrophage cell surface expression in relation to disease activity?
- 2) What is the effect of monocyte to macrophage differentiation on cell surface IL7 expression and the presence of sIL7. I would argue that the CD14+ cells in the SF of AS/PsA patients are macrophages not monocytes. Is the increase of IL7R or sIL7R due to macrophage differentiation or in vivo activation. Comparing monocytes and in vitro monocyte differentiated macrophages would address this question.
- 3) What is the basis for defining clusters 1, 2 and 3 in Figure 5c? they are not as clear to me as to the authors.
- 4) In Figure 4f it would seem that CCL5/LTA/MYB and TCF7 are not among the LPS/IL7 differentially expressed genes. the cells with this genotype seem to be both LPS and LPS/IL7.
- 5) the interpretation of Figure 5f is not clearly presented.
- 6) the authors suggest that DDX39A is associated with reduced sIL7R. An experiment to mechanistically support this association would strength their argument.
- 7) In one sentence Figure 2e I believe is referred to as 3e and in another Figure 3d is referred to as 1c.

Reviewer #2 (IL7 signalling, genetic mouse models, ILC)(Remarks to the Author):

The paper of Al-Mossawi et al. reports on the regulation of expression of cell surface and soluble IL-7R on human monocytes. It is demonstrated that an IL-7 allele, which is associated with several autoimmune disorders, determines csIL7R and sIL7R on LPS-stimulated monocytes but not across lymphocyte subsets or resting monocytes. sIL7R is produced almost exclusively by monocytes and this is associated with the splicing factor DDX39A. TNF stimulates enhances the expression of IL7R on monocytes. IL-7R+ monocytes were found in synovial fluid of patients suffering Spondyloarthritis. The analysis is of high quality involving a high number of data sets. The impact of this paper could be increased if the authors would make attempts to get more information on the functional significance of sIL-7R and the functional characteristics of IL-7R+ monocytes in synovial fluid of SA patients.

Specific comments:

To confirm that DDX39A is responsible for the splicing of IL-7R, the authors may overexpress this gene in T cells. The prediction is that this would result in generation of sIL-7R

No attempts were made to isolate the IL-7R+ monocytes of Spondyloarthritis patients to confirm the sequence data that these are truly monocytes and to get information on the functional characteristics of these cells. This is important because the data in figure 5e show that the IL-7R+ monocytes express a bunch of genes that are expressed not only in T cells but also on innate lymphoid cells.

These include CD7 highly expressed on lymphocytes and much less on myeloid cells, the transcription factors TCF7 RORA BCL11B which are all required for development of ILC subsets, CD3E and D that can also be expressed on ILCs. LTB is highly expressed in ILC as well. One can also say that these cells are ILC that happen to express CD14

The authors don't take into consideration that IL-7Ra is also a component of TSLP receptor complex. Do the IL-7R+ monocytes express the TSLP receptor?

Reviewer #3 (Autoimmunity, IBD, systems immunology)(Remarks to the Author):

Al-Mossawi and colleagues describe that functional IL7R and a soluble splice variant is expressed on monocytes after LPS stimulation and that a SNP associated with several autoimmune disease influences the relative distribution of cell surface and soluble receptors. These data are convincing. Based on these data, the authors propose that the SNP predisposes for autoimmune disease by regulating IL7R expression in monocytes. The authors then go one to demonstrate that IL7R expression is functional.

Specific comments

1. While the data on IL7R expression in monocytes are convincing, also because unstimulated monocytes do not express IL7R, the IL7R data on T cells require a more careful interpretation. Absence of IL7R, as used by the author is not a reliable read-out system, because naïve cells are nearly 100% positive, negativity is mainly determined by the frequencies of effector T cells. Upon stimulation, a complete loss within 24 hours cannot be expected, certainly not in non-proliferating cells, and if found would likely be due to subset selection rather than transcriptional regulation. To decide whether transcriptional changes are observed, mean fluorescence intensity would need to be assessed. Overall, the experimental system and read-out systems are not optimal to study IL7R expression on T cells. T cell receptor activation induces IL7R loss due to FOXO degradation and should therefore not induce the cell surface nor the soluble form. These limitations do not influence the conclusions on monocytes, which is the major finding of this paper, however, the conclusion that T cells do not produce a soluble form of the IL7R is not supported.
2. While TNF may contribute to IL7R expression, its contribution to IL7R variability is small.
3. Blocking studies with infliximab require a control antibody to exclude non-specific effects.
4. Does TNF induce IL7R as well as sIL7R?
5. In the t-SNE analysis in Figure 4, the separate cluster of IL7R/sIL7R+ cells appears to be CD14 positive. The main cluster, where IL7R and sIL7R appear to be at least in part more discordant, should be discussed. Since the emphasis of the manuscript is on the co-expression of IL7R and sIL7R, this is obviously of importance.
6. It would be of interest to see single cell RNA data on IL7 responsiveness in monocytes differing for the SNP.
7. More genes are downregulated than induced, and this is not always clearly stated. Functional consequences therefore remain obscure. In the introduction, it is stated that TCF1/7 is induced while it is downregulated.
8. AS is associated with the SNP, while RA is not. Was there a rationale why the authors studied RA and AS? Was there a difference?

Reviewer #1 (Autoimmunity, innate immunity, cytokine signalling)(Remarks to the Author):

This manuscript describes the presence of IL7R on monocytes after TLR stimulation, which is mediated in part by TNF. The increase appears greater in those with the rs932555 CC allele or the rs6897932 TT alleles, which are in LD. LPS induced DDX39A mRNA expression is related to the sIL7R. Differentially expressed genes are present following IL7 stimulation of LPS treated monocytes showing functionality of the expressed IL7 receptor. Single cell RNAseq is presented for control monocytes and AS/PsA monocytes and their synovial fluid counterparts. Addressing the following issues would strengthen the manuscript.

1) which allele of rs6897932 is associated with AS and MS? what is the proposed relation of sIL7 vs monocyte/macrophage cell surface expression in relation to disease activity?

We thank the reviewer for their comments, the C allele of rs6897932 is disease associated with MS, whilst in AS, the peak association is with rs11742270 (Cortes et al, Nature genetics 2013). rs11742270 is in complete linkage disequilibrium with rs6897932. Given rs6897932 is a coding SNP and has a characterised functional role in splicing, we have focused our efforts on this SNP. We have now made this clear in the text (page 2, line 19).

From a mechanistic perspective, we demonstrate that carriers of rs6897932 disease-associated allele generate increased monocyte derived sIL7R upon exposure to inflammatory stimuli. sIL7R has been shown to prolong IL-7 half-life (Lundstrom, PNAS 2013) and in turn promotes T cell-mediated inflammation. Consistent with this mechanism, AS patients homozygous for the disease associated C allele have a higher CD8 lymphocyte count (Cortes et al, Nature genetics 2013).

2) What is the effect of monocyte to macrophage differentiation on cell surface IL7 expression and the presence of sIL7. I would argue that the CD14+ cells in the SF of AS/PsA patients are macrophages not monocytes. Is the increase of IL7R or sIL7R due to macrophage differentiation or in vivo activation. Comparing monocytes and in vitro monocyte differentiated macrophages would address this question.

We agree with the reviewer that determining the exact state of differentiation of synovial CD14+ cells from monocyte to macrophage is important. Our data do not support the hypothesis that CD14+ IL7R+ cells represent macrophages however.

Firstly, with respect to the *ex vivo* monocyte stimulation assays, marked induction of IL7R transcript is noted by 2h (as per Figure 3a), whilst surface expression is maximal by 24 hours, a point much in advance of completion of monocyte to macrophage differentiation (which typically takes > 1 week). To further explore the effect of prolonged LPS on monocyte surface IL7R we performed time-courses of surface expression (see below) from two individuals. Notably, we find IL7R surface staining falls post 24h of stimulation. In addition, we have carried out further single cell RT qPCR on in-vitro LPS stimulated monocytes (updated figure 4) and we

observe similar CD163 expression on the IL7R positive subset, arguing against their differentiation towards an inflammatory macrophage phenotype.

Finally, single-cell data from the joint shows that the IL7R+ cluster has lower expression of macrophage differentiation markers CD163, MS4A7 and CD16 (FCGR3A) compared to the other CD14+ SFMC clusters (see figure below).

Taken together, this additional data would suggest IL7R expression occurs on a subset of monocytes at a stage well in advance of macrophage differentiation.

3) What is the basis for defining clusters 1, 2 and 3 in Figure 5c? they are not as clear to me as to the authors.

Figure 5c depicts the independently derived clusters which were defined using a K-means based unsupervised algorithm that takes into account the most variable genes in the dataset. These are depicted in figure 5f, and distance apart on a t-sne does not necessarily represent degree of dissimilarity. This approach is comparable with other published work in this field using the same Seurat R package (Villani et al Science 2017). We now provide a new supplementary table 4 which provides the gene expression data for each of the clusters identified in the analysis.

4) In Figure 4f it would seem that CCL5/LTA/MYB and TCF7 are not among the LPS/IL7 differentially expressed genes. the cells with this genotype seem to be both LPS and LPS/IL7.

The reviewer has raised a good point and in our previous single cell qPCR analysis we were unable to dissect genetic effects. To explore such effects further we have now increased the sample size to 10 individuals genotyped at the rs6897932 SNP (ratio 4:6 CC:TT). Our updated Figure 4 shows a clear effect of genotype on soluble versus membrane expression of IL7R.

We have made significant changes to the manuscript in light of these further experiments as detailed in the new figures 4d and 4e which show the t-SNE of co-expression across this larger population of cells and the effect of genotype on sIL7R and IL7Rmb at the single cell level. We have edited the manuscript (page 9-10, lines 12-1) to read :

To determine whether individual cells might co-express IL7Rmb and sIL7R, whilst taking into account genotype at rs6897932, we performed single cell quantitative PCR (qPCR) on monocytes (n=1818) isolated from 10 individuals who were homozygous at rs6897932 (4:TT, 6CC). We amplified sIL7R and full-length IL7R, as well as transcripts selected on response to LPS or IL-7, including at baseline and post-exposure to LPS or LPS+IL-7. Applying t-distributed stochastic neighbour embedding (t-SNE) algorithm to these data revealed that a subset of monocytes strongly expressed IL7Rmb post-LPS. Notably, these same cells were responsible for expression of sIL7R and co-expressed CCL5 and MALAT1 (Figure 4d). Using this further cohort of individuals we saw a strong genetic effect of rs6897932 on the distribution of expression of both IL7Rmb and sIL7R expression across cells, with more cells expressing sIL7R and at a higher level in individuals homozygous for the C allele (Figure 4e). Finally, when we compared cells post-LPS to those post LPS+IL-7

(24h) we were able to observe a genotype-specific effect of treatment with IL-7 on expression of MALAT1 (Figure 4f).

5) the interpretation of Figure 5f is not clearly presented.

We're sorry that the figure was difficult to interpret. We have improved the presentation of the figure firstly by increasing the resolution of the heatmap and secondly by clarifying in the text (page 10, lines 11-19) that the expression in the heatmap relates to the clusters identified in the tSNE plot in 5c.

6) the authors suggest that DDX39A is associated with reduced sIL7R. An experiment to mechanistically support this association would strength their argument.

We show a clear unbiased and highly statistically significant correlation from primary cells but agree this does not show cause causation. However, DDX39A plays a central role in mRNA splicing and export of many genes (Yamazaki *et al* Mol Biol Cell. 2010 Aug 15;21(16):2953-65.) and thus over-expression will inevitably lead to a number of off target effects and would not give a clear answer. An ideal experiment that would obviate such confounding observations would be to use genetic determinants of DDX39A expression as instrumental variables in an Mendelian Randomization approach – however, the effect size of these in available datasets means this is not currently possible and is outside the scope of this study.

7) In one sentence Figure 2e I believe is referred to as 3e and in another Figure 3d is referred to as 1c.

We are grateful to the reviewer for their attention to detail and have now corrected this.

Reviewer #2 (IL7 signalling, genetic mouse models, ILC)(Remarks to the Author):

The paper of Al-Mossawi et al. reports on the regulation of expression of cell surface and soluble IL-7R on human monocytes. It is demonstrated that an IL-7 allele, which is associated with several autoimmune disorders, determines csIL7R and sIL7R on LPS-stimulated monocytes but not across lymphocyte subsets or resting monocytes. sIL7R is produced almost exclusively by monocytes and this is associated with the slicing factor DDX39A. TNF stimulates enhances the expression of IL7R on monocytes. IL-7R+ monocytes were found in synovial fluid of patients suffering Spondyloarthritis. The analysis is of high quality involving a high number of data sets. The impact of this paper could be increased is the authors would make attempts to get more information on the functional significance of sIL-7R and the functional characteristics of IL-7R+ monocytes in synovial fluid of SA patients.

Specific comments:

To confirm that DDX39A is responsible for the splicing of IL-7R, the authors may

overexpress this gene in T cells. The prediction is that this would result in generation of sIL-7R

We thank the reviewer for this comment. While we agree that the effect of DDX39A on splicing in T cells may be of interest, our manuscript is focused upon the comparatively unexplored expression of IL7R in monocytes. Using an unbiased approach in a large expression dataset we make the observation that expression of DDX39A is anti-correlated with independently measured protein sIL7R from the same monocytes. The role of DDX39A in T-cells is outside the scope of this manuscript.

No attempts were made to isolate the IL-7R⁺ monocytes of Spondyloarthritis patients to confirm the sequence data that these are truly monocytes and to get information on the functional characteristics of these cells. This is important because the data in figure 5e show that the IL-7R⁺ monocytes express a bunch of genes that are expressed not only in T cells but also on innate lymphoid cells. These include CD7 highly expressed on lymphocytes and much less on myeloid cells, the transcription factors TCF7 RORA BCL11B which are all required for development of ILC subsets, CD3E and D that can also be expressed on ILCs. LTB is highly expressed in ILC as well. One can also say that these cells are ILC that happen to express CD14.

We thank the reviewer for this observation and are in agreement that the gene expression profile identified in IL7R⁺ monocyte population from synovial fluid has overlapping genes with ILCs. However, these cells also are high expressers of *LYZ* as well as *CD14*, both are canonical myeloid genes. Indeed, human transcriptomic data for ILCs shows they do not express *CD14* or *LYZ* (Bjorklund et al., Nature Medicine 2016). Moreover, the definition of ILCs dictates that they should be negative for lineage markers including CD14 and so we think this observation in itself excludes an ILC identity. However, to further confirm these observations, we have proceeded to sort CD11b⁺ CD14⁺IL7R⁺ and CD14⁺IL7R⁻ control cells from the synovial fluid of SpA patients. With these we demonstrate:

- i) These cells have a myeloid morphology on forward and side scatter flow cytometry parameters and are discrete from the lymphoid gate.
- ii) Using qPCR we confirm expression of both full length and sIL7R, and observe no difference in CD14 or LYZ expression in the IL7R⁺ subset. Moreover, we do see consistently higher expression of TCF7, LTB and BCL11B in the IL7R⁺ monocyte subset. These data are now included in Supplementary Figure 8.

The authors don't take into consideration that IL-7Ra is also a component of TSLP receptor complex. Do the IL-7R⁺ monocytes express the TSLP receptor?

This is an important point that we mentioned in the original manuscript (Page 13, lines 22-23 Discussion):

"IL7R can also dimerize with the thymic stromal lymphopoietin receptor^{23,24}, however monocyte expression of this is very low and it is not induced by LPS⁵"

We have further looked at *TSLP* expression in our single cell data and similarly are unable to find expression of this (see response to reviewer 1 point 2).

Reviewer #3 (Autoimmunity, IBD, systems immunology)(Remarks to the Author):

Al-Mossawi and colleagues describe that functional IL7R and a soluble splice variant is expressed on monocytes after LPS stimulation and that a SNP associated with several autoimmune disease influences the relative distribution of cell surface and soluble receptors. These data are convincing. Based on these data, the authors propose that the SNP predisposes for autoimmune disease by regulating IL7R expression in monocytes. The authors then go one to demonstrate that IL7R expression is functional.
Specific comments

1. While the data on IL7R expression in monocytes are convincing, also because unstimulated monocytes do not express IL7R, the IL7R data on T cells require a more careful interpretation. Absence of IL7R, as used by the author is not a reliable read-out system, because naïve cells are nearly 100% positive, negativity is mainly determined by the frequencies of effector T cells. Upon stimulation, a complete loss within 24 hours cannot be expected, certainly not in non-proliferating cells, and if found would likely be due to subset selection rather than transcriptional regulation. To decide whether transcriptional changes are observed, mean fluorescence intensity would need to be assessed. Overall, the experimental system and read-out systems are not optimal to study IL7R expression on T cells. T cell receptor activation induces IL7R loss due to FOXO degradation and should therefore not induce the cell surface nor the soluble form. These limitations do not influence the conclusions on monocytes, which is the major finding of this paper, however, the conclusion that T cells do not produce a soluble form of the IL7R is not supported.

We thank the reviewer for the comments. The focus this study is the novel finding of monocyte expression of IL7R post stimulation and the genotypic basis of this. To contextualise this with reference to the presumed T cell source of most sIL7R, we performed experiments to measure sIL7R protein secretion on a per cell basis, exploring the effect of divergent activators and at baseline. It is notable that across none of these conditions did the level of sIL7R from T-cells approach that from stimulated monocytes. It maybe that a separate stimulus is required for release of T cell sIL7R, but exploration of whether this is the case is outside the scope of this study. What we can conclude from our data however is that activated monocytes show more sIL7R production compared to either resting or activated T cells on a per cell basis, but we agree that our data does not exclude T cell production of sIL7R. We have now clearly stated this in the text to ensure our conclusions are not overstated. (Page 12, lines 2-6).

Whilst T cells cannot be excluded as a significant source of sIL7R, we find that activated monocytes produce markedly greater sIL7R than either resting or activated T cells and this

occurs in a *rs6897932*-delineated manner, suggesting that in the context of innate immune activity, monocytes can significantly contribute to the local pool of sIL7R.

2. While TNF may contribute to IL7R expression, its contribution to IL7R variability is small

The review is correct that we observe a more limited effect of TNF on IL7R expression than LPS, suggesting other cytokines may be involved in this response. Nonetheless, we find in experimenter blinded assays that the effect of TNF again is genotype specific and this reaches statistical significance.

3. Blocking studies with infliximab require a control antibody to exclude non-specific effects.

Many thanks for highlighting this and we agree that an isotype control is needed. We have now carried out additional experiments using a human IgG1-kappa isotype control (Biolegend cat number 403501) and show that the isotype does not modulate LPS-induced surface IL7R compared to Infliximab. In addition, we have performed an additional control using a chimeric human antibody against CD20 (Rituximab) and again show that this does not interfere with LPS induction of IL7R unlike infliximab. Representative flow cytometry plots are presented below.

4. Does TNF induce IL7R as well as sIL7R?

The reviewer raises an interesting and valid point. To explore this, we have carried out additional experiments exposing isolated monocytes to TNF. These show that TNF can indeed induce a sIL7R, albeit at a lower level than LPS. This data is in

keeping with the surface effect we see with TNF and is now included in figure 3 (panel e)

5. In the t-SNE analysis in Figure 4, the separate cluster of IL7R/sIL7R+ cells appears to be CD14 positive. The main cluster, where IL7R and sIL7R appear to be at least in part more discordant, should be discussed. Since the emphasis of the manuscript is on the co-expression of IL7R and sIL7R, this is obviously of importance.

The reviewer raises an important point and we believe that discordancy in CD14 and IL7R expression in the previous experiment may reflect the comparatively few cells analysed. To address this and further explore IL7R expression at the single cell level we have performed additional experiments across cells from further genotyped donors. This experiment now has data from an increased number of cells and 10 genotyped donors (CC:TT, 6:4). In these new data we no longer observe significant discordancy between IL7R and CD14 expression, although we are now able to see a clear effect of genotype on soluble vs membrane bound IL7R expression (Figure 4d,e).

6. It would be of interest to see single cell RNA data on IL7 responsiveness in monocytes differing for the SNP.

We thank the reviewer for this interesting suggestion. We have now carried out this analysis in new single-cell qPCR analysis from 10 genotyped individuals. Previously we found MALAT1 to be one of the most significantly IL-7 modulated genes in the bulk sequencing data. In this further data we are now able to observe a genotype-specific effect of IL-7 on MALAT1 expression, with divergent responses to IL-7 that are not observed with LPS alone. This result is now included in figure 4 panel f.

7. More genes are downregulated than induced, and this is not always clearly stated. Functional consequences therefore remain obscure. In the introduction, it is stated that TCF1/7 is induced while it is downregulated.

We apologise for any confusion over this point. The IL7R+ monocyte subset is characterised as having high expression of TCF7, whereas TCF7 is similarly modulated by IL-7, although in the short exposure it was downregulated. To make this point clearer we have edited the introduction such that it reads (line 18-20 page 3):

Exposure of LPS stimulated monocytes to IL-7 is associated with a specific transcriptional profile and this corresponds to the genes differentially expressed in the synovial IL7R+ monocyte subset, characterized by high expression of TCF7 and CCL5.

8. AS is associated with the SNP, while RA is not. Was there a rationale why the authors studied RA and AS? Was there a difference?

We apologise to the reviewer for the confusion. We have not studied RA samples in this paper and all patients included had either peripheral, or axial spondyloarthritis.

Reviewers' comments:

Reviewer #1 (Remarks to the Author):

The authors have made efforts to address the concerns of the reviewers. In my opinion the responses have not been fully satisfactory.

1) The authors continue to argue that the cells in the inflammatory synovial fluids are monocytes. The data they provide, clearly supports the fact that the cells are different from circulating monocytes. Further, in mice where more detailed studies are possible, monocytes were present in the bone marrow, circulation and spleen (Science 325: 612, 2009; Front. Immunol 5: 514, 2014). Data from another inflammatory arthritis are informative. The cells in the synovial fluid come through the tissue, before entering the synovial fluid, and the cells in the tissues are macrophages, not monocytes (Nat Reviews Rheum 12: 472, 2016). Additionally, macrophages have been described in RA synovial fluid, which are very different from matched peripheral blood (Arthritis Rheum 56: 2192, 2007). The new data presented is not convincing that the cells in the synovial fluid are monocytes.

2) When asked to determine of macrophage differentiation vs activation was responsible for the IL-7R changes, a time course for the expression of IL-7R in response to LPS was done. This does not address the question. Following adherence of monocytes to plastic, changes are observed at 24 hours. Human monocytes have been differentiated into macrophages in the presence of M-CSF at 4 or 7 days (J. Immunol 177:7303, 2006; Am. J Path 170:2007). In my opinion this should be done.

3) One control for the infliximab experiment is insufficient.

4) When asked for a mechanistic experiment to determine the role of DDX39A in the expression of sIL7R, the authors argue that addressing this is outside the scope of the study. Monocytes and in-vitro differentiated macrophages can be transfected to reduce the expression of a variety of molecules to determine the effects.

Reviewer #2 (Remarks to the Author):

The paper has been improved. That CD14+IL-7R+ cannot be ILCs because they express CD14 is a cycle reasoning. Mjosberg (as well as all other ILC researchers) use anti CD14 to exclude CD14+ cells from their ILC preparations. That the forward and side scatter of is "myeloid" is also not a strong argument for not considering these cells as ILCs. However, this reviewer realises that further more detailed characterisation of these cells is beyond the scope of the paper. That the TSLP receptor is undetectable in single cell analysis is not an argument that TSLP as the transcript expression may be below the detection limit. In fact TSLP induces CD80 in human CD14+ cells from peripheral blood (DOI: 10.1007/s00011-011-0310-0). So it is well possible that sIL-7 can affect TSLP signalling. The authors may discuss this.

Reviewer #3 (Remarks to the Author):

The revisions have addressed my concerns. The TNF blocking data included in the cover letter are not very impressive because the LPS-induced IL7R is much less than that shown in Figure 1a and the antibody shows a shift in the MFI of the entire population, however, it is reassuring that this shift is not seen with the control antibodies. In general, primary data such as this figure should be included in the supplement. As a minor point, some of the y-axis labels are not clearly described, e.g., what is shown in Figure 1f, MFI or percent positive cells?

Reviewers' comments:

Reviewer #1 (Remarks to the Author):

The authors have made efforts to address the concerns of the reviewers. In my opinion the responses have not been fully satisfactory.

1) The authors continue to argue that the cells in the inflammatory synovial fluids are monocytes. The data they provide, clearly supports the fact that the cells are different from circulating monocytes. Further, in mice where more detailed studies are possible, monocytes were present in the bone marrow, circulation and spleen (Science 325: 612, 2009; Front. Immunol 5: 514, 2014). Data from another inflammatory arthritis are informative. The cells in the synovial fluid come through the tissue, before entering the synovial fluid, and the cells in the tissues are macrophages, not monocytes (Nat Reviews Rheum 12: 472, 2016). Additionally, macrophages have been described in RA synovial fluid, which are very different from matched peripheral blood (Arthritis Rheum 56: 2192, 2007). The new data presented is not convincing that the cells in the synovial fluid are monocytes.

2) When asked to determine of macrophage differentiation vs activation was responsible for the IL-7R changes, a time course for the expression of IL-7R in response to LPS was done. This does not address the question. Following adherence of monocytes to plastic, changes are observed at 24 hours. Human monocytes have been differentiated into macrophages in the presence of M-CSF at 4 or 7 days (J. Immunol 177:7303, 2006; Am. J Path 170:2007). In my opinion this should be done.

We thank the reviewer for their observations regarding this interesting question. The induction of IL7R transcript in monocytes is rapid, with IL7R RNA expression being robustly detected within 2 hours. Moreover, we demonstrate IL-7R can be induced via divergent TLR ligands and TNF – therefore expression of IL-7R appears to be a generalized response to inflammatory stimuli and not in keeping with an early stage of differentiation.

It is difficult to further describe the properties of the myeloid cells in the joint beyond that at a single cell transcriptomic level. These IL7R expressing cells express key monocyte markers and have expression profiles consistent with a sub-population of monocytes previously described by Villani et al (Science 2016) - thus we are describing a recognized population of monocytes. The reviewer is correct however in that it is not known whether macrophages could similarly express IL7R, although it would be surprising if they similarly expressed the other markers of this subset.

To address the reviewer's question we have differentiated macrophages as advised. We find no induction of IL7R during the early differentiation (48h) and complete differentiation into macrophages is not associated with expression of IL7R (<1%). Interestingly, both day 2 and day 7 macrophages become refractory to LPS induction of IL7R suggesting that the IL7R response is specific to monocytes and strongly argues against the synovial fluid derived cells being macrophages. These new data are incorporated into a new Supplementary Figure 11 and are referred to on page 11, line 1:

"Finally, to explore whether the observed IL7R⁺ myeloid cells might be macrophages, we performed a standard macrophage differentiation assay in primary monocytes from three individuals. We found no induction of IL7R expression however, and interestingly these cells became refractory to LPS induced IL7R upregulation (Supplementary Figure 11)."

3) One control for the infliximab experiment is insufficient.

We acknowledge this and have repeated control IgG experiments (n=6), and find the reduction in LPS induced monocytes IL7R expression is not a general property of immunoglobulins. These data are added to supplementary figure 2.

4) When asked for a mechanistic experiment to determine the role of DDX39A in the expression of sIL7R, the authors argue that addressing this is outside the scope of the study. Monocytes and in-vitro differentiated macrophages can be transfected to reduce the expression of a variety of molecules to determine the effects.

Our data shows a highly significant, agnostically determined, genetic association between the expression of DDX39A and soluble IL7R protein from the same monocytes obtained from 161 randomly assayed samples ($P=5.1^{-10}$) – indeed DDX39A was the most significantly associated coding gene on the array. The cells are primary and have not been subjected to interference. DDX39A encodes a putative RNA helicase that plays a key role in the splicing of 100s-1000s genes and is implicated in a myriad of cellular processes. It has been shown that interfering with its expression will thus interfere with many pathways. Given this, we believe it likely that the large effects of modulating DDX39A expression would preclude any insights into the effect of splicing on this single induced transcript. Importantly, it would not be possible to rule out interference in the monocyte LPS response that would interfere with the induction of IL7R initially. Finally, given the very large effect of genotype, any experiment could not be carried out in one individual, but would require many samples from each homozygous allele – not an insignificant request when adding in the variable responses between individuals to

siRNAs. Finally, we have shown at the reviewers' request the absence of IL7R expression in macrophages, whilst the expression of IL7R in monocytes is relatively transient. Thus, the experiment proposed would require considerable additional preparative work and it would be unclear what timeframe to analyse the cells. Moreover, the observation of association of sIL7R and expression of DDX39A is a notable but not key finding of this study and thus the suggested work is outside the scope of this study which is primarily a human primary genetic analysis.

Reviewer #2 (Remarks to the Author):

The paper has been improved. That CD14+IL-7R+ cannot be ILCs because they express CD14 is a cycle reasoning. Mjosberg (as well as all other ILC researchers) use anti CD14 to exclude CD14+ cells from their ILC preparations. That the forward and side scatter of is "myeloid" is also not a strong argument for not considering these cells as ILCs. However, this reviewer realises that further more detailed characterisation of these cells is beyond the scope of the paper. That the TSLP receptor is undetectable in single cell analysis is not an argument that TSLP as the transcript expression may be below the detection limit. In fact TSLP induces CD80 in human CD14+ cells from peripheral blood (DOI: 10.1007/s00011-011-0310-0). So it is well possible that sIL-7 can affect TSLP signalling. The authors may discuss this.

We are grateful to the reviewer for their further time and attention, and are pleased that they find the study has been improved.

Whilst we feel the evidence supporting an ILC nature of these cells (which, due to the number of cells involved, cannot be supported in the bulk genotypic analysis of isolated monocytes responding to LPS) is lacking, we concur that further resolution of ILC IL7R expression profiles and any potential overlap is outside the scope of this current work.

As per the reviewer, we have questioned the possibility that the induced IL7R is interacting with the TSLP receptor. In addition to expression of the absent expression of TSLP receptor in single cell sequencing, we similarly find in analysis of the previous array data from our original eQTL study and also the bulk RNA sequencing we report here, there is no detectable expression of this gene in monocytes however. Thus we think the likelihood that IL-7 is affecting TSLP signaling, at least in monocytes, is very low. Nonetheless, it is conceivable that the high monocyte-derived sIL7R might influence TSLP signaling on other cells. Whilst we show in this revision that macrophages do not express IL7R, a role for the IL7R on other myeloid cell such as eosinophils is not precluded, but this is outside the scope of this work. We have added the following sentence to the paper:

IL7R can also dimerize with the thymic stromal lymphopoietin (TSLP) receptor^{23,24}, however monocyte expression of this is very low and it is not induced by LPS⁵, thus we focused this study on investigating the role of IL-7 on monocyte biology. An additional role for sIL7R in the modulation of TSLP signaling cannot be excluded – indeed the UK biobank data shows an association of this allele with eosinophil count²¹.

Reviewer #3 (Remarks to the Author):

The revisions have addressed my concerns. The TNF blocking data included in the cover letter are not very impressive because the LPS-induced IL7R is much less than that shown in Figure 1a and the antibody shows a shift in the MFI of the entire population, however, it is reassuring that this shift is not seen with the control antibodies. In general, primary data such as this figure should be included in the supplement. As a minor point, some of the y-axis labels are not clearly described, e.g., what is shown in Figure 1f, MFI or percent positive cells?

We are pleased to have been able to address the concerns of the reviewer. We concur that the TNF blocking data shown were not impressive – but this partly illustrates the large degree of variance in the IL7R response in monocytes. We agree that this should be in the supplement and have moved this figure to Supplementary Figure 2, adding an additional boxplot showing the absence of IgG control response in a further 6 samples. With respect to Figure 1f., this demonstrates RNA expression showing a correlation between 2h TNF expression and 24h IL7R expression. The legend has been edited for clarity and now reads:

1 f) Array derived RNA expression of TNF at 2h LPS assayed versus RNA expression of IL7R from monocytes from same individuals at 24h LPS.

REVIEWERS' COMMENTS:

Reviewer #1 (Remarks to the Author):

I have no additional comments.